

# Enabling BOINC in Infrastructure as a Service Cloud Systems

Diego Montes[1], Juan A. Añel[1,2], Tomás F. Pena[3], Peter Uhe[4,5], and David C. H. Wallom[5]

[1]EPhysLab, Universidade de Vigo, Ourense, 32004, Spain
[2]Smith School of Enterprise and the Environment, University of Oxford, Oxford, UK
[3]Centro de Investigación en Tecnoloxías da Información (CITIUS), University of Santiago de Compostela, Santiago de Compostela, 15782, Spain
[4]School of Geography and the Environment, University of Oxford, Oxford, UK
[5]Oxford e-Research Centre, University of Oxford, OX1 3QG, UK.

*Correspondence to:* Diego Montes (kabute@uvigo.es)

**Abstract.** Volunteer or Crowd computing is becoming increasingly popular to solve complex research problems, from an increasing diverse range of areas. The majority of these have been built using the Berkeley Open Infrastructure for Network Computing (BOINC) platform, which provides a range of different services to manage all computation aspects of a project. The BOINC system is ideal in those cases where not only does the research community involved need low cost access to massive
computing resource but also that there is a significant public interest in the research done.

We discuss the way in which Cloud services can help BOINC based projects to deliver results in a fast, on demand manner. This is difficult to achieve using volunteers, and at the same time, using scalable cloud resources for short on demand projects can optimize the use of the available resources. We show how this design can be used as an efficient distributed computing platform within the Cloud, and outline new approaches that could open up new possibilities in this field, using climate*prediction*.net
as a case study.

**Keywords.** BOINC, Cloud, CPDN, Volunteer computing

## 1 Introduction

Traditionally, climate models have been run using supercomputers because of their vast computational complexity and high cost. Since its early development, climate modelling has been an undertaking that has tested the limits of High-Performance
Computing (HPC). This application of models to answer different types of questions has led to their being used in manners not originally foreseen. This is because, for some types of simulations, it can take several months to finish a modelling experiment given the scale of resources involved. One reason for including climate modelling as an High-Throughput Computing (HTC), as opposed to an HPC problem is due to the application design model, where there is a number (not usually greater than twenty) of uncoupled, long-running tasks, each corresponding to a single climate simulation and its results.
The aim of increasing the total number of members in an ensemble of climate simulations, together with the need to achieve increased computational power to better represent the physical and chemical processes being modelled, has been well understood for some decades in meteorological and climate research. Climate models make use of ensemble means to improve the





accuracy of the results and quantify uncertainty, but the number of members in each ensemble tends to be small due to computational constraints. The overwhelming majority of research projects use ensembles that generally contain only a very small number of simulations, which has an obvious impact in terms of the statistical uncertainty of the results.

The climate*prediction*.net project (CPDN) was created in 1999 (Allen, 1999; CPDN, 2015) as a distributed computing
initiative to address the uncertainties described above. Its aim is to run thousands of different climate modelling simulations in order to research the uncertainties associated with some of the parameters. This is essential for understanding how small changes or variations in initial conditions can affect both the models themselves and the results of climate simulations. The project is currently run by the University of Oxford using volunteer computing via the BOINC (Berkeley Open Infrastructure for Network Computing) framework (BOINC, 2014; Anderson, 2004). In its early use of distributed computing, CPDN became
a precursor of the Many-Task Computing (MTC) paradigm (Raicu et al., 2008).

CPDN has been running for more than 10 years and faces a number of evolving challenges:

- an increasing and variable need for new computational and storage resources;

- the processing power and memory of current volunteers' computers, restricts the use of more complex models and higher resolution.

- the need to manage costs and budgeting. This is of particular interest in researching on-demand projects requested by external research collaborators and stakeholders;

To address these issues we have explored the combination of MTC/volunteer and Cloud computing as a possible improvement or, extension to, a real existing project. This kind of solution has previously been proposed for scientific purposes by Iosup et al. (2011) and is supported by initiatives such as Microsoft Azure for Research (Microsoft, 2014).

## 2   Background

It is not the aim of this paper to describe the internals of BOINC, and for better comprehension of the problem that we are trying to solve, it is recommended to review previous works about this knowledge such as Ries et al. (2011).

### 2.1   Problem Description

Here we describe some of the problems that we intend to address, as well as propose implementations of possible solutions:

- to run more complex and computationally more expensive versions of the model, resources greater than volunteer computers can provide may be needed. One solution is a re-engineering and deployment of the client side from a volunteer computing architecture to an Infrastructure as a Service (IaaS) based on Cloud Computing (e.g. Amazon Web Services, AWS);

- there is a growing need for an on-demand and more predictable return of simulation results. A good example of this
is urgent simulations for critical events in real-time (e.g., floods) where it is not possible to rely on volunteers; instead



a widely available and massive scaling system is preferable (like the one described here). The current architecture and infrastructure based on BOINC does not provide a solution that can be scaled up for this purpose. This is because the models are running over a heterogeneous and decentralized environment (on a number of variable and different volunteers' computers with varying configurations), where their behaviour cannot be clearly anticipated or measured, and any control over the available resources is severely limited;

– a rationalization of the costs is required (and establishing useful metrics), not just for internal control but also to provide monetary quotations to project partners and funding bodies; this led us to the need of the development of a Control Plane together with a frontend to display the statistics information and metrics;

– use of Free Software in order to promote scientific reproducibility (Añel, 2011);

– complete documentation of the process, allowing knowledge to be transferred or migrated easily to other systems (Montes, 2014). Additional explanations can be found in the appendices (Appendix A, Appendix B and Appendix C).

Furthermore, in this work we wish to prove the feasibility of running complex applications in this environment. We use weather@home (Massey et al., 2015), a high resolution regional climate model nested in a global climate model as an example. The remainder of this paper is organized as follows. We firstly present benchmarks of the weather@home application run in AWS, in section 3.1, then describe the migration of the CPDN infrastructure to AWS in section 3.2. We also describe our control plane in section 3.3 to conduct the simulations and manage the cloud resources. Lastly the results are discussed in the conclusion.

## 3   BOINC deployment to the Cloud

### 3.1   Application benchmarks in Amazon Web Services (AWS)

The example presented here is running CPDN in AWS. AWS is the largest Infrastructure as a Service (IaaS) provider, it is very well documented, as well as being the most suitable solution for the problem at present (and with fewer limitations than other providers).

The first step was to benchmark different AWS EC2 instance types[1] to determine their performance running CPDN simulations. These tests were done with a range of instance types, but only choosing instance types that have HMV virtualization available. EBS gp2 storage[2] was used for all instances for ease of comparison. These tests were carried out running multiple copies of a single workunit, in parallel, with the number of simulations matching the number of vCPUs (hyperthreads) available to each instance type. For each instance types at least 4 tests were run.

For benchmarking purposes, short 1 day climate simulations were run. The model used here is weather@home2 (atmosphere only model HadAM3P (Gordon et al., 2000), driving the regional version of the same model, HadRM3P (Pope et al., 2000)).

---

[1]https://aws.amazon.com/ec2/instance-types/
[2]http://docs.aws.amazon.com/AWSEC2/latest/UserGuide/EBSVolumeTypes.html





This version of the model uses the MOSES 2 land surface scheme. The region chosen is at 0.22 degree ($\approx$ 25 km) resolution over Europe.

Figure 1a shows the average time to run all of the simulations on a particular instance, by instance type. We see a general trend of smaller instances performing better than larger instances. This is likely due to the hardware these instances are on being at a lower load. Running only a single simulation per instance resulted in similar run times for instances of the same category (e.g. c4), and they are not shown in the figure. However, we have verified that it is more cost effective to run the maximum number of simulations per instance than to run instances at a lower load.

Figure 1b shows the estimated cost of running a one year simulation on each each instance type. The pricing here is based on the Spot Price[3] in the cheapest availability zone in the us-east-1 region as of June 2016. This shows that the current generation compute optimized instances (c4) had three out of the four most cost effective choices, but other small instance types are amongst the cheapest. We emphasise that these results are very variable in time and between regions. In the us-west-1 region and us-west-2 regions, the cheapest instance types were m4.large and m4.xlarge respectively, due to the lower spot price for those particular instances in those regions.

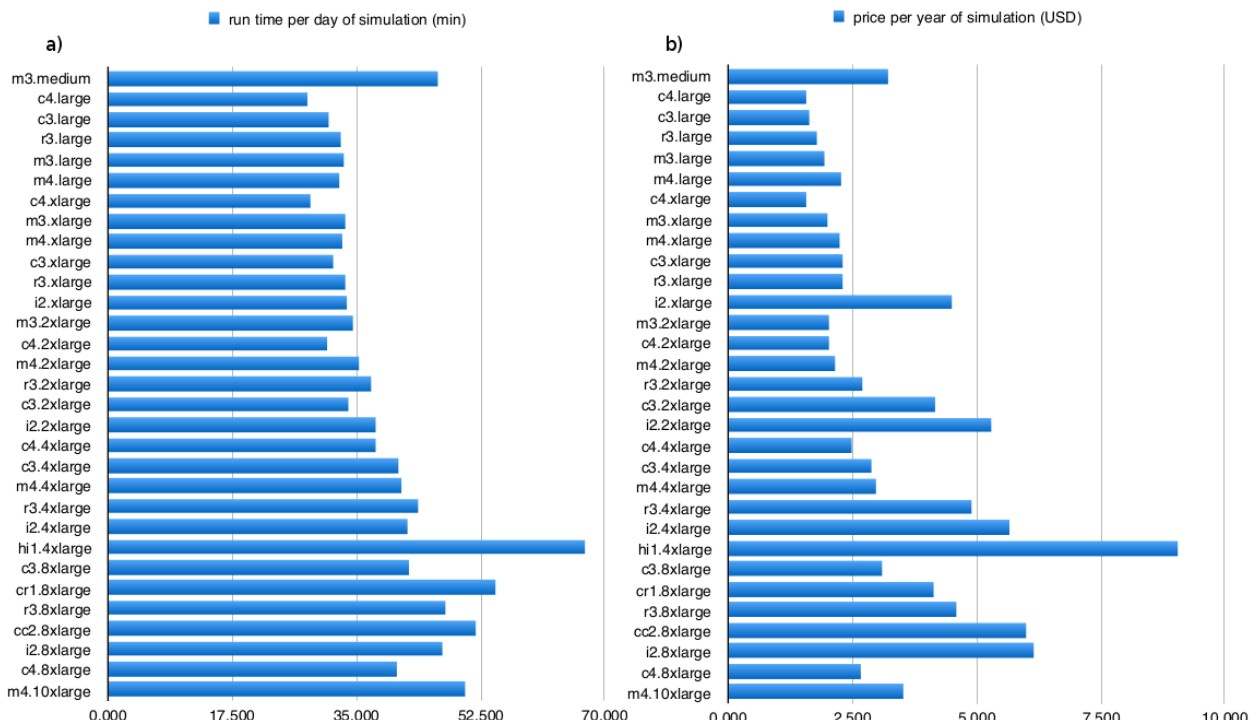

**Figure 1.** a) Workunit run time and b) Cost per simulation year.

---

[3]https://aws.amazon.com/ec2/spot/pricing/



## 3.2 CPDN infrastructure in AWS

Based on the previous tests, new infrastructure was designed on the cloud (Figure 2). Several steps were required for its implementation, as described below.

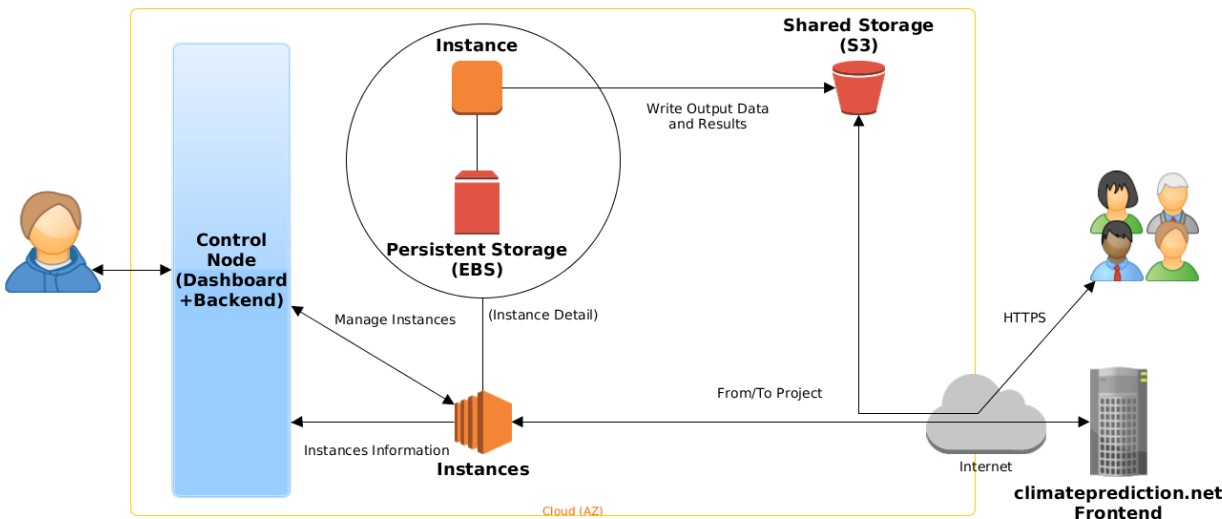

**Figure 2.** Proposed Cloud Infrastructure

### 3.2.1 Computing Infrastructure

1. First of all a template was created to allow automated instance creation including:

- instance selection, based on the benchmarks presented in section 3.1;

- base operating system installation: Amazon Linux (64 bit) was used for this work;

- storage definition: 16 GB, persistent, standard EBS for this case;

- firewall configuration: inbound only SSH(22) accepted, outbound everything accepted;

- installation and configuration of BOINC client inside the template image, including its dependencies such as 32 bit libraries. It is recommended to use the latest version from git;

2. instance contextualization: post installation configuration, for example in AWS this is achieved by creating a machine image (AMI) and adjusting it by selecting the appropriate options such as the Kernel image (AKI);

3. (optional) installation and configuration of AWS EC2 Command Line interface. This can be useful to debug or trou-
bleshoot issues with the infrastructure;



### 3.2.2 Storage Infrastructure

Another problem that needs to be solved is the need for a decentralized, low-latency and world-wide accessible storage for the output data (each simulation (36000 workunits) generates ∼656 GB of results). A solution for this could be a distributed (accessed within different and synchronized world-wide endpoints) and scalable massive storage (Figure 3). Here we tested an

5 architecture in which the clients send the results (tasks) to an Amazon Simple Storage Service (S3) bucket (storage endpoint). At the same time, CPDN can access these data over the internet to run postprocessing (e.g., a custom assimilator); this can be achieved using the AWS API.

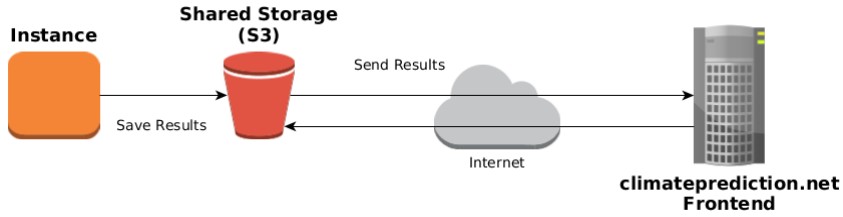

**Figure 3.** Shared Storage Architecture

### 3.3 Project Control Plane

Having set up the computing and storage infrastructure, we still lack a control plane to provide a layer for abstraction and

10 automation, and providing more consistency to the project. The aim of developing the Central Control System (Figure 4) is to provide a Cloud-agnostic, easy-to-use frontend (Figure 5) to manage the experiments with minimal knowledge of the underlying architecture and obtain a real-time overview of the current status (including the resources used and run completion data). Moreover, the Central Control System lends more consistency to the view of the project as an IaaS by providing a simple interface (both backend and frontend).

The Control Plane is still in its early developmental stages (e.g., although it is cloud-agnostic, so far only AWS has a connector and is supported), and further work will describe its improvements over time.

It consists of two main components:

– backend: this provides the user with a RESTful API with basic functionalities related to simulation information and management, with the intention of providing (even more) agnostic access to the Cloud;

– frontend: this makes it easier to communicate with the API as intuitively and simplistically as possible;

The core component, the RESTful backend (using JSON), provides simple access and wraps common actions: start simulation with n-nodes, stop simulation, modify simulation parameters (n-nodes), get simulation status, and get simulation metrics.



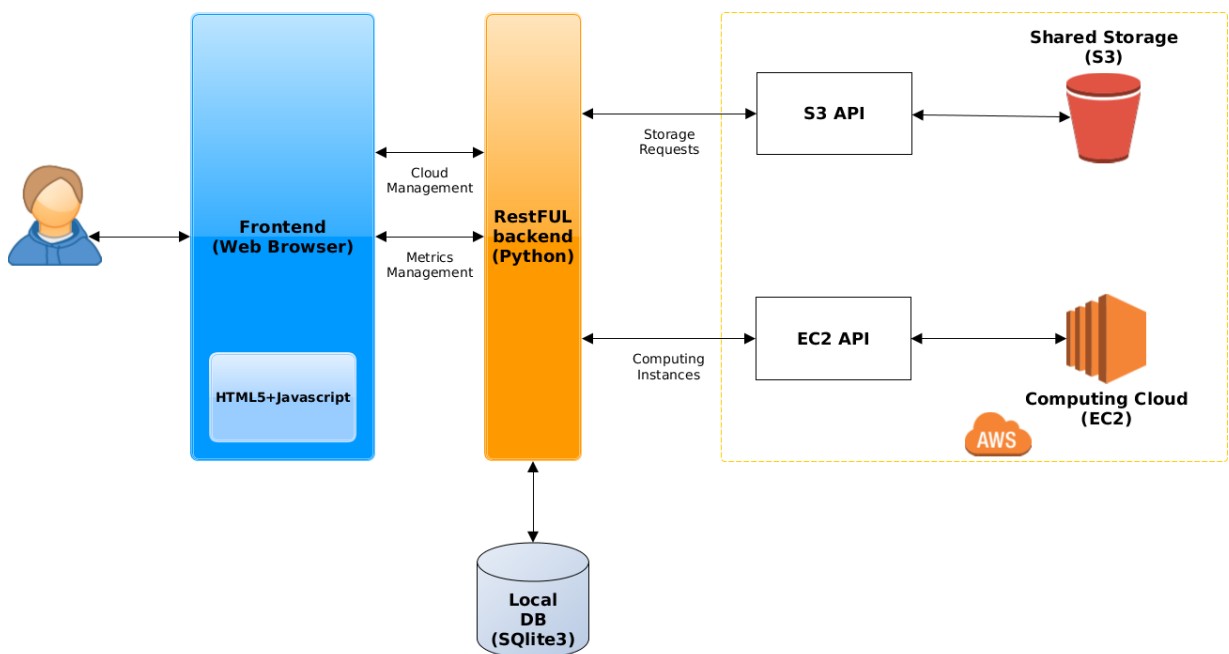

**Figure 4.** Dashboard and metrics application architecture

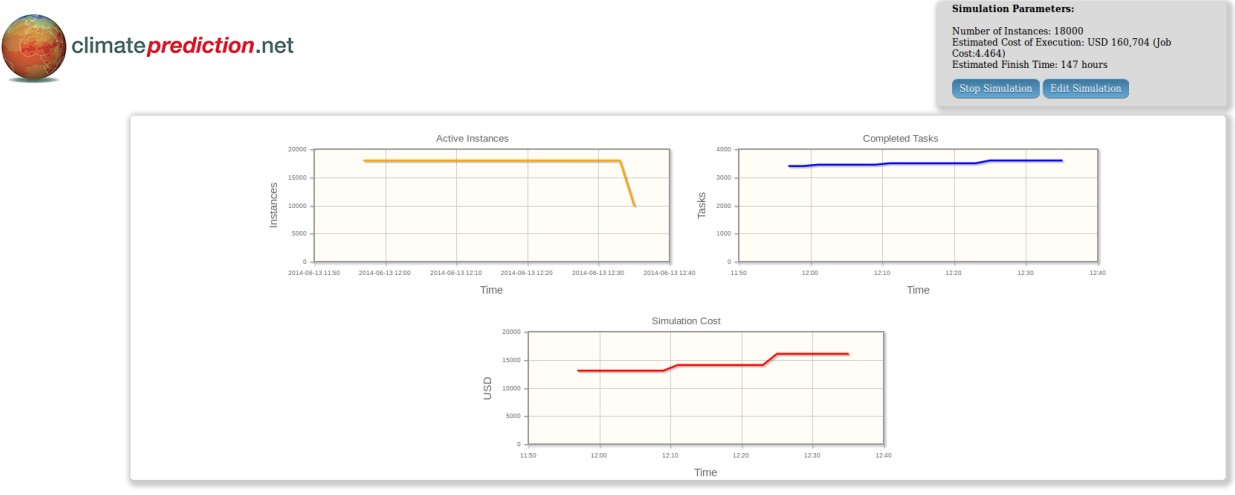

**Figure 5.** Dashboard





## 4    Conclusions

Several experiments (using all the defined infrastructure) were done by using standard workunits developed by the climatepre-diction.net/weatherhome project. We processed workunits from two main experiments: the weatherhome UK floods (Schaller et al., 2016) and the weatherhome Australia New Zealand project (Black et al., 2016), both with an horizontal resolution of 50 km.

It has been successfully demonstrated that it is possible to run simulations of a climatic model using infrastructure in the Cloud; while this might not seem complex, to the best of our knowledge it has never previously been tested. This efficient use of MTC resources for scientific computing has previously been used to facilitate real research in other areas (Añel et al., 2014; Schaller et al., 2014).

We have benchmarked a number of Amazon EC2 instance types running CPDN workunits. Prices for spot instances vary significantly over time and between instance, but we estimate a price as low as $1.50 to run a one year simulation based on the c4.large instance in the us-west-1 region in June 2016 (see Figure 1). To optimise the costs of running simulations in this environment it will be important to automatically re-evaluate the spot prices to choose the cheapest instance type at the time simulations are submitted.

It is interesting to note that Cloud services enable us to achieve a given number of tasks completed in some cases five times faster than using the regular volunteer computing infrastructure. However, the financial implications can only be justified for critical cases where stakeholders are able to jusity through a specific cost benefit analysis.

This research has also served as a basis for obtaining new research funding as part of climateprediction.net for state-of-art studies using Cloud Computing technologies. This project is based on demonstrated successes in the application of technologies

and solutions of the type described here.

In summary, the achieved high-level objectives were:

– the client side was successfully migrated to the Cloud (EC2);

– the upload server capability was configured to be redirected to AWS S3 buckets;

– different simulations were successfully run over the new infrastructure;

25

– a Control Plane (including a Dashboard: frontend and backend) was developed, deployed and tested;

– a comprehensive costing of the project and the simulation were obtained, together with metrics;

Future improvements should focus on providing more logic to the interaction with client status (such as through RPC calls) allowing more metrics to be pulled from them, and creating new Software as a Service (a SaaS layer). From the infrastructure point of view, two main improvements are possible: first, a probe/dummy automated execution will be needed to adjust the

30

price to a real one before each simulation; second, full migration of the server side into the Cloud, allowing the costs of data transfer and latency to be dramatically reduced.



## 5   Code Availablity

Minor code pieces and automations were used to perform this work. They are fully described in this paper. They do not avoid

5   the reproducibility of the presented work. They can be shared under request.

Author contributions. All the authors participated into the design of the experiments and the analysis of results. D. Montes implemented the full infrastructure experiments. P. Uhe carried out the benchmarking. All authors participated in the writing of this paper.

10   *Acknowledgements.* We thank Andy Bowery, Jonathan Miller and Neil R. Massey for all their help and assistance with the internals and specifics of the CPDN BOINC implementation. The compute resources for this project were provided under the AWS Cloud Credits for Research Program.





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

## Appendix A: Computing Infrastructure Design and Implementation

The new computing infrastructure was built over virtualized instances (AWS EC2). Amazon provides also Autoscaling Groups that allow the user to define policies to add or remove dynamically instances triggered by a defined metric or alarm. As the purposes of this work are to use the rationalization of the resources and to have full control over them (via the Central System), as well as any type of Load Balancing or Failover, this feature will not be used in the Cloud side but in the control system node that serves as backend for the Dashboard.

After tasks have been setup in the server side and are ready to be sent to the clients (this can be currently checked into the public URL http://climateapps2.oerc.ox.ac.uk/cpdnboinc/server_status.html), the new workflow for a project/model execution is:

1. The (project) administrator user configures and launches a new simulation via the Dashboard.

2. The required number of instances are created based on a given template that contains a parametrized image of GNU/Linux with a configured BOINC client.

3. Every instance connects to the server and fetchs 2 tasks (1 per CPU, as the used instances have 2 CPU).

4. When a task is processed, the data will be returned to the server, and also stored into a Shared Storage so it will be accessible for a given set of authorized users.

5. Once there are not tasks available, the Control Node will shut down the instances.

It should be noted that, at any point, the administrator will be able to have real time data about the execution (metrics, costs...) as well as change the running parameters and apply them over the infrastructure.

### A1  Template Instance Creation

In order to be able to create an homogeneous infrastructure the first step is to create an (EC2) instance that can be used as template for the other ones.

The high level steps to follow to get a Template Instance (with the parameters defined in the Table 1) are exposed below:




| OS Image (AMI): | Amazon Linux AMI 2014.03.1 (64 bit) |
|---|---|
| Instance Type: | m1.large |
| Firewall (Security Groups): | Inbound: Only SSH (22) Accepted |
| | Outbound: Everything Accepted. |
| Persistent Storage: | Root 16GB (volume type: standard) |

**Table 1.** Parameter for the template instances

1. On the AWS dashboard click "Launch Instance", then select the given OS image (AMI) type.

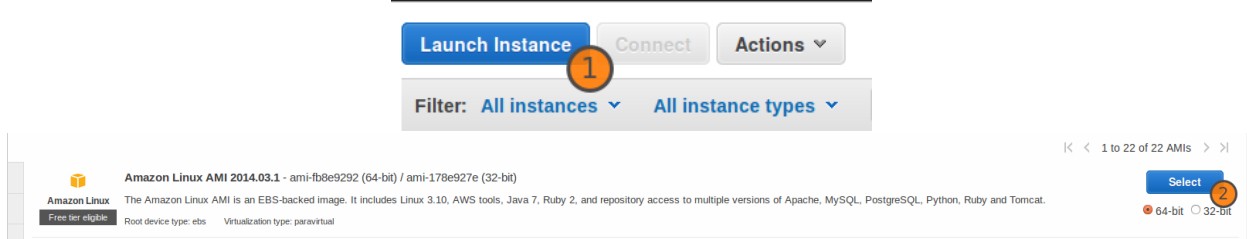

2. Select the image type .

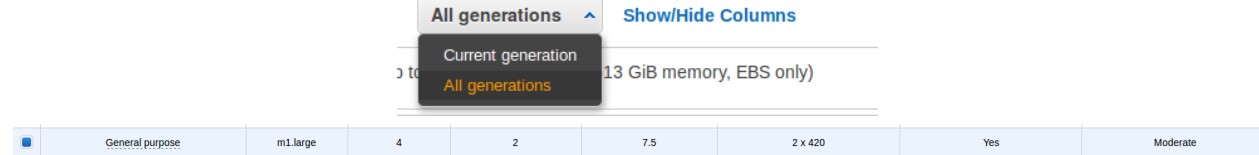

3. Revise and set the parameters:

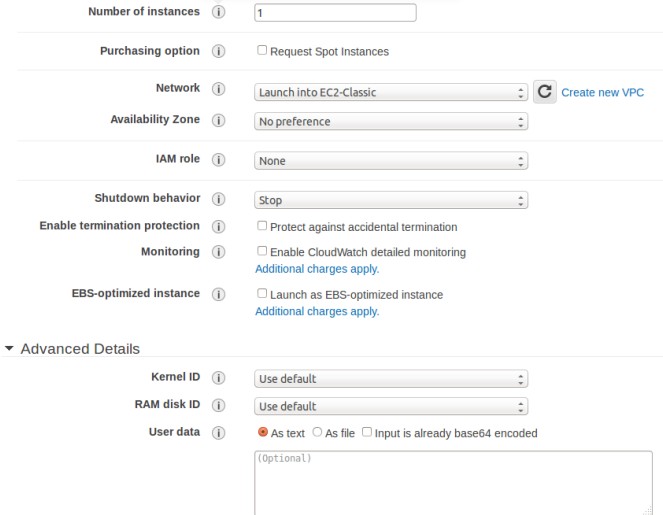





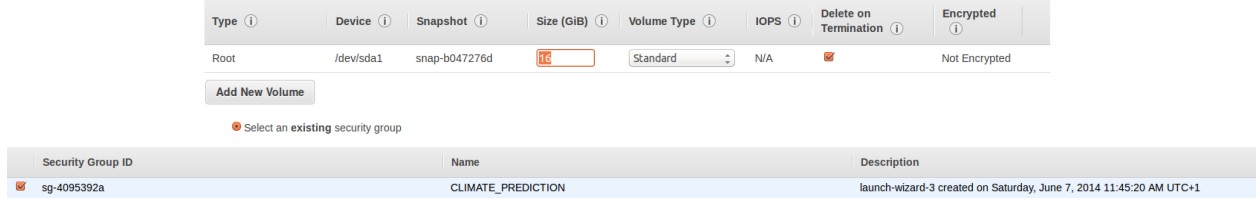

4. Launch the Template Instance.

Note: remember to create a new keypair (public-private key used for passwordless SSH access to the instances) and save it

(it will be used for the Central System), or use another one that already exists and is currently accessible. Because of the limited space into this article the line length (new line) has been truncated with \, please consider this when running any command described in here.

### A1.1  Installing and testing AWS and EC2 Command-Line Interface

(Prerequisites: wget, unzip and Python 2.7.x)

This step is optional, but it is highly recommendable because this will be advanced control of the infrastructure through the shell. The follow description applies and have been tested on Ubuntu 14.04 Canonical Ltd. (2014), but can be reproduced into any GNU/Linux system:

First, create an *Access Key* (and secret/password), via the AWS web interface in the *Security Credentials* section. With this data the *AWS_ACCESS_KEY* and *AWS_SECRET_KEY* variables should be exported/updated, please have in mind that this

mechanism will be also used for the Dashboard/Metrics Application:

```
$ echo "export AWS_ACCESS_KEY = \
<your-aws-access-key-id>" \
>> $HOME/.bashrc
$ echo "export AWS_SECRET_KEY = \
<your-aws-secret-key>" \
>> $HOME/.bashrc

$ source $HOME/.bashrc

#Download and install the AWS CLI
$ wget https://s3.amazonaws.com/\
aws-cli/awscli-bundle.zip &&\
unzip awscli-bundle.zip &&\
sudo ./awscli-bundle/install -i \
/usr/local/aws -b /usr/local/bin/aws
```



```
#Download EC2 API tools
wget http://s3.amazonaws.com/\
ec2-downloads/ec2-api-tools.zip &&\
```

```
sudo mkdir /usr/local/ec2 &&\
sudo unzip ec2-api-tools.zip -d \
/usr/local/ec2
```

```
#Remember to update the
#PATH=$PATH:/usr/local/ec2/\
#ec2-api-tools-<API_VERSION>/bin
$ echo "export JAVA_HOME=/usr/lib/jvm\
/java-6-openjdk-amd64/" >>
```
15
```
$HOME/.bashrc
```

```
$ source $HOME/.bashrc
```

### A1.2    Installing BOINC and its dependencies

The project executes both 32 and 64 bits binaries for the simulation so once the Template Instance is running, the needed

20 packages and dependencies need to be installed via:

```
$ sudo yum install expat.i686 flac.i686 \
fontconfig.i686 freetype.i686 gamin.i686 \
glib2.i686 glibc.i686gnutls.i686 \
gtk2.i686 libX11.i686 libXau.i686 \
```
25
```
libXext.i686 libXfixes.i686  libXft.i686 \
libXi.i686 libcom_err.i686 libgcc.i686 \
libgcrypt.i686 libgpg-error.i686 \
libstdc++.i686 libxcb.i686 xcb-util.i686 \
zlib.i686 libcurl.i686 openssl097a.i686
```

30    The version of BOINC used will be the latest from git (Torvalds, 2014). To download and compile it:

```
#Install needed development tools
$ sudo yum groupinstall 'Development Tools'
```



```
   $ sudo yum install git libcurl \
   libcurl-devel openssl097a.i686 \
   openssl-devel
 5
   #Fetch BOINC source code, compile and
   #install the client
   $ git clone \
   git://boinc.berkeley.edu/boinc-v2.git \
boinc && cd boinc && ./_autosetup \
   && ./configure --disable-server \
   --disable-manager --enable-optimize \
   && make \
   && sudo make install
   #Create user and set permissions and
   #ownership
   #(in case that we want a boinc user)
   $ sudo adduser boinc
20 $ sudo chown boinc /var/lib/boinc
```

Once the BOINC client is installed it must be configured so it will automatically run on every instance with the same parameters:

1. Create a new account in the project:

```
$ boinccmd --create_account \
      climateprediction.net <EMAIL> \
      <PASSWORD> <NAME>
```

2. With the account created (or if already done) the client needs to be associated to the project by creating a configuration file with the user token:

```
$ boinccmd --lookup_account \
      climateprediction.net <EMAIL> \
      <PASSWORD>| grep "account key" \
```





```
     | sed 's/\(.*\): \(.*\)/\2/g' \
     | xargs boinccmd --project_attach \
     climateprediction.net
```

```
     #Status check
     $ boinccmd --get_state
```

3. Make BOINC to start with the system (ec2-user will be used because of permissions):

```
     $sudo echo 'su - ec2-user -c \
     "cd /home/ec2-user/boinc-client/\
10   bin/ && ./boinc --daemon"'\
     >> /etc/rc.local
```

### A1.3 Simulation Terminator

An essential piece of software, developed for this work, is the *Simulation Terminator*, that decides if a node should shutdown itself in case that workunits were not processed for a given amount of time (by default 6 hours, via cron) or there are no jobs
15 waiting on the server.

This application will be provided upon request to the authors.

To install it (by default into */opt/climateprediction/* ):

```
$ sudo ./installClient
>> Simulation Client succesfully installed!
```

20 When an instance is powered off, it will be terminated (destroyed) by the *Reaper* service, that runs into the Central Control System.

### A2 Contextualization

Now that the Template Instance is ready, this is that all the parameters have been configured and the BOINC client is ready to start processing tasks, the next stage is to contextualize it. This means that a OS image will be created from it, which will
25 give our infrastructure the capacity of being scalable by creating new instances from this new image. Unfortunately this part is strongly related with the Cloud type, and although can be replicated into another systems, by now it will only explicitly work in this way for AWS.

The steps to follow:

1. On the instances list (AWS Dashboard), select the Template Instance, right click and select *Create Image*, name of
30   the image: *CLIMATE_PREDICTION_TEMPLATE*. This will create a disk Image that can be used for a full Instance Template (AMI):



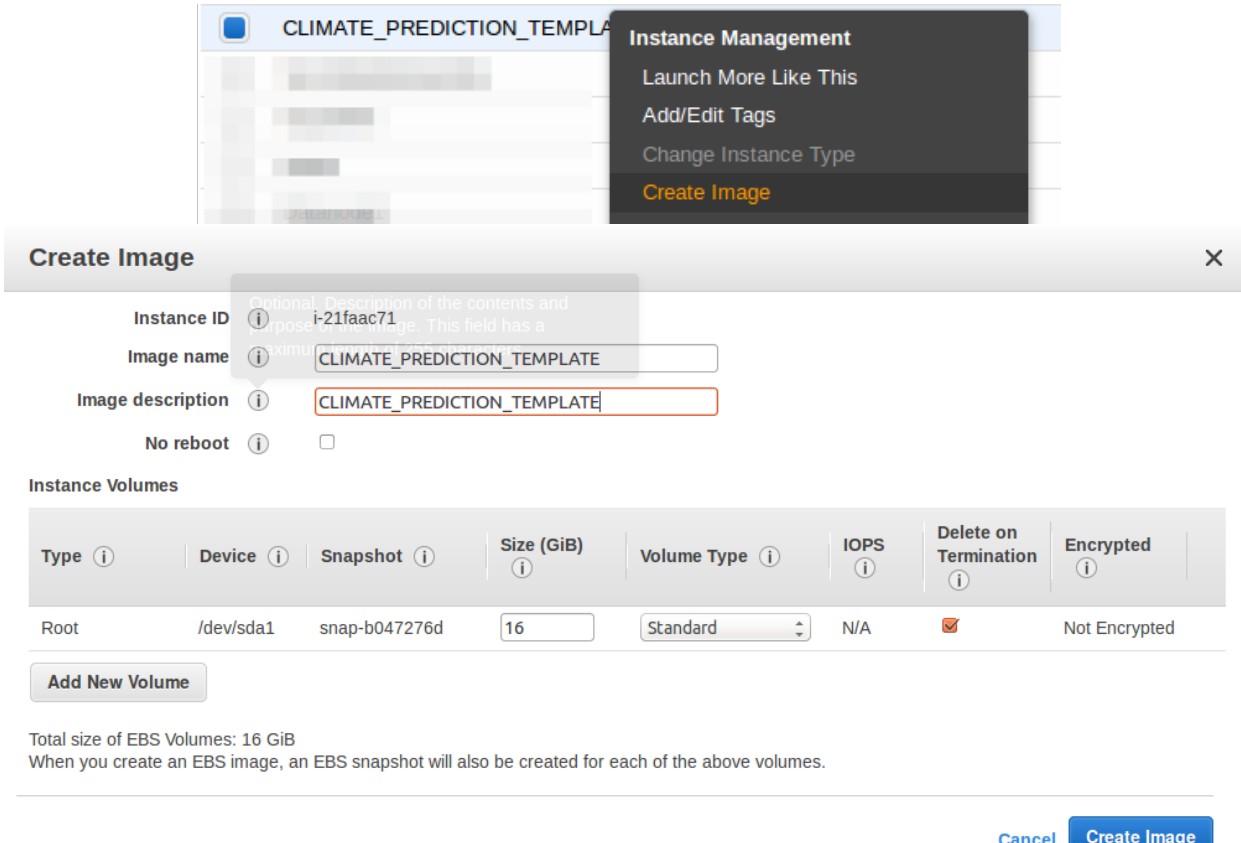

2. To finally create the Instance Image (in the AWS Dashboard) go to Images→AMIs and right click on *CLIMATE_PREDIC-TION_TEMPLATE* and fill the parameters, at least name as *CLIMATE_PREDICTION_TEMPLATE* (the same as Image, for better identification) and match the kernel image (AKI) with the original Template Instance (currently: aki-919dcaf8). This step is very important, otherwise the new instances created for the project simulations won't boot correctly.

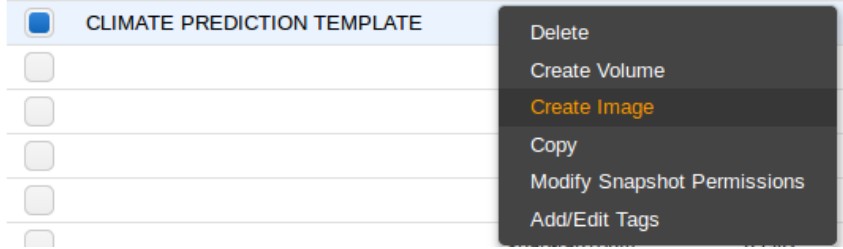





At this point the Computing infrastructure is ready to be deployed and scaled, this will be done trough the Dashboard.

## Appendix B: Upload Server

Once a client has processed a workunit, the task (result) is created and sent to the defined Upload Server, that for the CPDN
5   is http://cpdn-upload2.oerc.ox.ac.uk/cpdn_cgi/file_upload_handler. This needs to be done in a transparent way for the clients
and without modifying the server because we don't want to affect the actual running experiments (but in the future the servers
should distribute a configuration that directly points to the S3 bucket). To do this the data should be intercepted, and this can
be done in 2 steps/components:

  – **The name resolution** should be *faked* by changing the CNAME http://cpdn-upload2.oerc.ox.ac.uk/cpdn_cgi/file_upload_
10     handler point to the created S3 bucket endpoint. Bind documentation can be reviewed for this in howtoforge.com (2010)

  – **A web server as endpoint**, with HTTP and HTTPs support, configured to resolve the URL http://$UPLOAD_SERVER/
    cpdn_cgi/ (the jobs are created to target this URL). To simplify this stage, the storage provided by AWS, S3, will be used
    because it has a simple HTTP(s) server that supports all the required HTTP methods (GET and POST). The (expected)
    content of the *file_upload_handler* must be:

15  ```
    <data_server_reply>
          <status>1</status>
    </data_server_reply>
    ```

  1. Access to the S3 Service from the AWS dashboard.

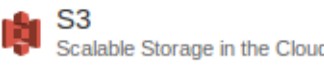





2. Click on "Create" Bucket, the name should be *CLIMATE_PREDICTION* and must be in the same Region than the instances.

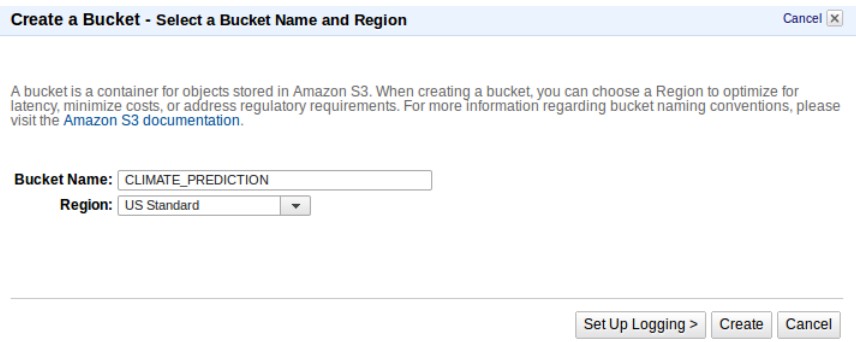

3. Activate (in the Options) the HTTP/HTTPs server.

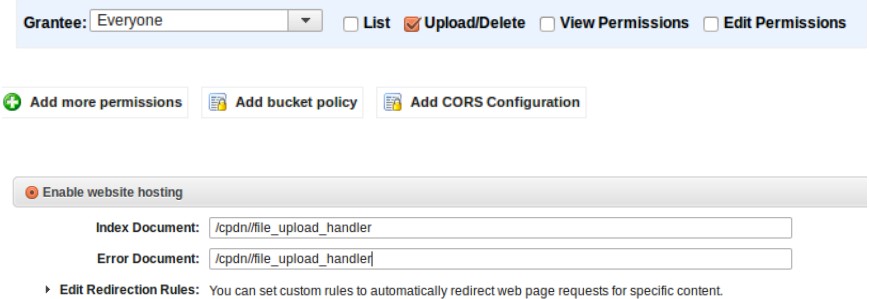

4. To secure the bucket remember to modify the policy so only allowed IP ranges can access it (in this case only IP ranges from instances and from CPND servers).

```
    {
10      "Version": "2012-10-17",
        "Id": "S3PolicyId1",
         "Statement": [
            {
            "Sid": "IPAllow",
15          "Effect": "Allow",
            "Principal": "*",
            "Action": "s3:*",
            "Resource":
              "arn:aws:s3:::CLIMATE_PREDICTION/*",
20
            "Condition" : {
```



```
        "IpAddress" : {
          "aws:SourceIp": "ALLOWED_IP_RANGES"
                        }
                      }
5         } ]
      }
```

## Appendix C:  Central Control System and Dashboard

### C1    Backend and API

The backend of the Central System consists into:

– a *RESTful (Representational state transfer) API* over Flask (a Python web microframework (Grinberg, 2013)) that controls the Infrastructure (with Boto, a Python interface to Amazon Web Services (Garnaat, 2010)).

– A *Simple Scheduler*, that will be in the background and will take care that the simulation is running with the given parameters (e.g. all the required instances are up).

– The *Reaper*, a subsystem of the Simple Scheduler that is some sort of garbage collector and will terminate powered off instances in order to release resources.

### C1.1   API Reference

The backend can be reused and integrated into another systems in order to give the full abstraction over the project. The available requests (HTTP) are:

– **Get Simulation Status:**

   Call: *status*

   Request Type *GET*

   Returns: *JSON* object with current simulation status:

```
{
 requestedInstances:
  <int:REQUESTED_INSTANCES>

 executionCost:
  <float:EXECUTION_COST_IN_USD>,
```



```
      workunitCost:
        <float:WORKUNIT_COST_IN_USD>,
      }
```

- **Get Metric:**

  Call: *metric/METRICNAME*

  Request Type *GET*

  Returns: *JSON* object with metric (time series):

  ```
      {
        <METRIC_VARIABLE>:
          [{<int|float:METRIC_DATA>,
             <timestamp:TIMESTAMP>}],
      }
  ```

- **Set/Modify Simulation:**

  Call: *simulation*

  Request Type *POST*

  Input: *JSON* object with simulation parameters, if already running will modify it:

  Returns: *JSON* object with result (0=fail, 1=succesful):

  ```
      {
        instances: <int:REQUESTED_INSTANCES>
      }

      {
        result: <0|1>
      }
  ```

- **Stop Simulation:**

  Call: *simulation/stop*

  Request Type *GET*

  Returns: *JSON* object with result (0=fail, 1=succesful):

  ```
      {
        result: <0|1>
      }
  ```



A simplistic (but functional) GUI (Graphical User Interface) has been designed to make it more understandable the execution of the simulation on the Cloud.

Two control actions are available:

- Start/Edit Simulation: sets the parameters (cloud type, number of instances...) of the simulation and runs it.

- Stop Simulation: forces all the instances to Terminate.

There are 3 default metrics (default time lapse: 6 hours):

- Active Instances: number of Active Instances.

- Completed Tasks: number of workunits successfully completed.

- Simulation Cost: accumulated Cost for the Simulation.

## C2   Installation and Configuration

The applications are intended to run at any GNU/Linux The only requirements are (apart from Python 2.7) Flask and Boto, that can be easily installed into any GNU/Linux :

```
$ pip install flask virtualenv \
  boto pysqlite daemonize
```

## C2.1   First configuration and Run

For this step the file `controlSystem.tar.gz`, which contains all the software and configurations for the Central System, needs to be uncompressed into /opt/climateprediction/, then just:

```
$ cd /opt/climateprediction
$ ./firstRun.py
>> Connector Type? aws
>> AWS Key: \
   <TYPE_YOUR_AWS_KEY>
>> AWS Secret: \
   <TYPE_YOUR_AWS_KEY_SECRET>

>> Dashboard Username: \
   <TYPE_DASHBOARD_USERNAME>
>> Dashboard Password: \
   <TYPE_DASHBOARD_PASSWORD>
```





```
[OK] Central System Ready to Run!
        Type ./run.py to start.
```

5 ```
#Start the service...
$ ./run.py
```

```
#... and Central System starts
#resolving backend and frontend requests
```

10    Optionally the configuration can be set manually by editing the file *Config.cfg* (parameters in <>):

```
#Main Configuration
[main]
connector=<CLOUD_CONNECTOR>
pollingTime=<REFRESH_TIME_IN_SECONDS>
```

```
[HTTPAuth]
user=<DASHBOARD_USERNAME>
password=<DASHBOARD_PASSWORD>
```

```
#AWS Credentials Configuration
[AWSCredentials]
KEY=<AWS_KEY>
PASSWORD=<AWS_SECRET>
```

```
#AWS Connector Configuration
[AWS]
AMI=<AMI>
instanceType=<INSTANCE_TYPE>
securityGroup=CLIMATE_PREDICTION
```
```
keyPair=<CLIMATE_PREDICTION_KEYPAIR>
```

## C3   Use and Project Deployment

Now that the Central System has been installed and configured, it will be listening and accepting connections into any network interface (0.0.0.0) on port 5000, protocol HTTP, so it can be accesed via web browser. Firefox or Chromium are recommended because of Javascript compatibility.

5   ### C3.1   Launch a New Simulation

When starting a simulation the number of instances will be 0. This can be changed by clicking "Edit Simulation", set the number into the input box and click on *Apply Changes*. Within some minutes (defined in the configuration file, in the *pollingTime* variable) the system will start to deploy instances (workers).

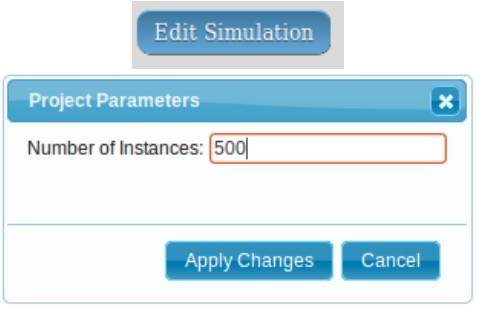

### C3.2   Modify a Simulation

If the number of instances needs to be adjusted when a simulation is running the procedure is the same than launching a new simulation (*Edit Simulation*). **Please be aware that if the number of instances is reduced unfinished workunits will be lost** (the scheduler will stop and terminate them using a FIFO).

15   ### C3.3   End Current Simulation

When a simulation wants to be stopped click *Stop Simulation*. This will reduce the number of instances to 0, copy the database as *SIMULATION-TIMESTAMP* for further analysis and reset all the parameters and metrics.