# Peer review of "Enabling BOINC in Infrastructure as a Service Cloud Systems"

_Geoscientific Model Development, 2016_

## Short Comment (SC1) · 21 Sep 2016

Thanks to the authors for an interesting paper. I found the discussion of the method and architecture they used informative. However, given that the authors have addressed *cost*, then they've missed an important component of the cost. They've estimated > 600 GB per simulation (is that per simulation-year), but they haven't factored the residence time in S3 into their cost. How long might they expect it to need to be resident, and what would that cost?

---

## Author Comment (AC1) · 29 Sep 2016

First of all thank you very much for taking the time to read our paper and to comment on it. We will add some more lines about data storage in the final reviewed version. We can note that the price of the storage did not have big impact into our simulations because AWS (and most of the public Cloud providers) have a low price for the space itself, offer free data transfer within the same region (which is actually the case for this paper) and it has the highest cost into transfers to the Internet (current S3 pricing for this can be found in here: https://aws.amazon.com/s3/pricing/ ). About how long is the data going to be stored: part of the purpose of our work was to be able to create massive simulations in the Cloud on-demand by using BOINC, so the results live just long enough for their analysis. Of course it seems interesting a future work about massive storage (e.g. terabytes for years), which could require other technologies like

[Figure]

Glacier and its integration with a CDN for better distribution.

The number that we give of around 600 GB per simulation is for a whole experiment, not per year simulated. We will clarify it in the reviewed version too.

Also some of the public cloud providers that we have been investigating have begun a move to waive egress charges (or some fraction of them if you read the small print) to academic users.

---

## Referee Comment (RC1) · Carlos Fernandez Sanchez (Referee) · 13 Oct 2016

The paper presents the implementation of a novel method for running ensemble climate simulations in the cloud. In particular, a connection of climateprediction.net project (CPDN) to Amazon Web Services (AWS) using the BOINC interface already in use and modifying it to use AWS. The paper is well structured and describes the scientific objectives and the implementation of the solution, and how to reproduce the results. More details about the benefits of using the cloud, apart from what is already known, would be expected, and also about how this solutions complements other computational solutions.

In particular, in my opinion it would be interesting to present more details about:

How the use of AWS compares against using other computational resources available (grid computing, supercomputers, clusters, domestic resources,...), in different aspects, including but not limited to the costs. For example the total time to get a full ensemble simulation including the pre and post-processing.

Compare the use of commercial cloud providers to other scientific cloud providers like EGI FedCloud

Could this approach be useful not only for Ensemble Predictions?

Include more details about the costs of moving and archiving the data. If every simulation generates 656 GB of data what is the costs of running many (ensemble) simulations? Include numbers

The paper includes enough good references (over 20)

Finally, no major typing errors were found in the document

―――――――――――――

---

## Referee Comment (RC2) · A. Arribas (Referee) · 24 Nov 2016

General Comment: This paper describes the use of AWS infrastructure to run BOINC as an alternative to its traditional application using volunteer computing. The paper is significant as it addresses a timely issue for many research projects that, because of public interest and/or financial constrains, have chosen volunteer computing until now. The increased availability and reduced costs of many cloud platforms makes them extremely attractive for research projects and this paper should be attractive to large number of readers.

The approach, quality and presentation of the paper is good. Particularly, I would like to thank the authors for their effort on the Appendices to provide the reader with a clear step-by-step guide to reproduce their work.

However, I believe the paper would benefit of further discussion of a series of points and I would like to encourage the authors to do so. Particularly, it would be extremely beneficial to have a better analysis of these points in the conclusions:

- Generating data is only one part of the story. Data needs to be stored and analysed. The costs of storage in the cloud can be substantial and the computational requirements for data analytics are very different to those needed to generate data in the first place, particularly if the data is to be open to a large number of scientists/data analysts

- Although I fully accept that this paper describes a proof of concept, the models run in this experiment are, relatively speaking, low resolution. There will be different challenges using higher resolution or more complex models. This could be related to the information shown in Figure 1 for run-time as it is surprising that smaller instances perform better than bigger instances. A better discussion of why this is the case would be very useful for other researchers as it has very clear financial implications.

Specific points: - Page 2, line 1: "the number of members in each ensemble tends to be small due to computational constrains" The use of computational resources in climate modelling is a balancing act between resolution, complexity and ensemble members as the authors point out a few lines earlier. It is not that the number of ensemble members is small due to computational constrains, it is often a conscious choice made by researchers. Having a big ensemble is desirable but it only makes sense if your model captures the right processes that you are interested in, otherwise you have the risk of just better sampling noise.

- Page 6, section 3.2.2 How could the data in S3 be used by other applications beyond climateprediction.net? How would this cope with much bigger datasets of hundreds of terabytes or petabytes?

A few typos have been detected. For example, in the opening pages: - page 1, line 15 should be: "... questions has led to them being ..." - page 1, line 17 should be: "... including climate modelling as a High ..." - page 2, line 24 should be: "... as well as

proposed ..." - page 3, line 27 should be: "... For each instance's type at least 4 test ..."

---

## Author Comment (AC2) · 23 Dec 2016

Thank you a lot for your review and comments. Answering your specific questions:

How the use of AWS compares against using other computational resources available (grid computing, supercomputers, clusters, domestic resources,...), in different aspects, including but not limited to the costs. For example the total time to get a full ensemble simulation including the pre and post-processing.

We have completed comparisons with the Oxford Supercomputer ARCUS-B (Uhe et al., Utilising Amazon Web Services to provide an on demand urgent computing facility for climateprediction.net, Proceedings of the 2016 IEEE 12th International Conference on e-Science), where it is shown that running the simulations on the supercomputer

Printer-friendly version

[Figure]

is more than twice as expensive than AWS spot instances but half the price that on-demand. Run times are similar, and this can be extrapolated to other supercomputers (of course depending on variables such as size, utilization and queueing on the resources). For our work we understand that "cluster" is an abstract and high level concept that include (and can be constructed on the top of) solutions over cloud computing. Compared with domestic resources (volunteers), it typically takes 7 times more (than AWS) in order to get an 80% of results returned while AWS returns a 100%.

Compare the use of commercial cloud providers to other scientific cloud providers like EGI FedCloud

A comparative of different cloud providers, is part of the aforementioned work and will be published in further papers but we would like to note that one of the coauthors of the paper (David Wallom) has participated actively in the design of the EGI FedCloud. Therefore, we are well aware of the advantages and limitations that it could have. The EGI Federated cloud is, as its name suggests, a federation of heterogeneous resource providers. So, they have all made different choices as to the infrastructure that they provide and the cloud management stacks they use to provide the user facing clouds services. Therefore we would actually only be testing any single system within that federation and so it would not be neither fair nor accurate to say that this is a performance benchmark of the 'EGI federated cloud'. We do also note though that due to its nature we may in future (if there are enough services providers with common hardware systems underpinning different choices in cloud management stack be able to test and compare performance of these different private cloud software solutions.

Could this approach be useful not only for Ensemble Predictions?

Our approach can be used for ensemble predictions as discussed in related works, but the main goal of this paper is to show that the model can be ran over the cloud and that BOINC can be a valid resource manager for any model. Sure, it could be used for

other problems in the field of climate research beyond ensemble studies

Include more details about the costs of moving and archiving the data. If every simulation generates 656 GB of data what is the costs of running many (ensemble) simulations? Include numbers

As mentioned, the cost of the data storage and transfer is non-existent for this work but have included specific numbers and prices in the final version of the paper (including S3 and Glacier). Also, it is worth to note that there is an AWS fee waiver for academic institutions that, usually, makes data transfer free.

---

## Author Comment (AC3) · 23 Dec 2016

We would like to thank the reviewer for the time and the effort to review our work and for the good and constructive comments. About the questions:

Generating data is only one part of the story. Data needs to be stored and analysed. The costs of storage in the cloud can be substantial and the computational require- ments for data analytics are very different to those needed to generate data in the first place, particularly if the data is to be open to a large number of scientists/data analysts

We totally agree with your comments but we would like to note that the impact of data transfer and storage (for us mainly S3) is minimal (and academic institutions can benefit

from waivers) Obviously this is completely dependent on the duration that you wish to store the complete raw results. Also within the scope of the submitted paper we are considering the execution of GCM type climate models and not the analysis of their outputs as the methods chosen are very different within different groups with no single clear method appearing as the community standard as yet. Anyway, we include more specific costs on the final version of the document.

Although I fully accept that this paper describes a proof of concept, the models run in this experiment are, relatively speaking, low resolution. There will be different challenges using higher resolution or more complex models. This could be related to the information shown in Figure 1 for run-time as it is surprising that smaller instances perform better than bigger instances. A better discussion of why this is the case would be very useful for other researchers as it has very clear financial implications.

We agree that there will indeed be different challenges for more complex modes. It is clear that models that require many processors to run will not benefit from the use of smaller instance types. However, the point we are making to highlight the need to accurately benchmarks on the application used to ensure that you are maximising the value of the computational resources that you are intending to use. The better performance with smaller instance is due to the fact that vCPUs are hyperthreads and in smaller instance types there is greater chance the CPU is running at a lower utilization and our instances can scavenge extra CPU cycles. This information is now included in the text (also see Uhe et al., Utilising Amazon Web Services to provide an on demand urgent computing facility for climateprediction.net, Proceedings of the 2016 IEEE 12th International Conference on e-Science).

Specific points: - Page 2, line 1: "the number of members in each ensemble tends to be small due to computational constrains" The use of computational resources in climate modelling is a balancing act between resolution, complexity and ensemble members as the authors point out a few lines earlier. It is not that the number of ensemble members is small due to computational constrains, it is often a conscious choice made

by researchers. Having a big ensemble is desirable but it only makes sense if your model captures the right processed"

The authors agree with this point and it is clear that the models resolution and the size of the ensemble is completely dependent on the experiments that the submitter is considering to analyse. It is though a common complaint of researchers that they cannot access the scale of resources desirable to create ensembles numbers large enough due to lack of resources or having to share them with other members of the community in larger national scale systems.

Re "only makes sense if your model captures the right processes that you are interested in" It should also be noted that if the model used doesn't capture the climate process necessary to be analysed for the application of interest then the researchers should be questioning why they are using that model in the first place. This is also again out of scope for the study being published here.

Page 6, section 3.2.2 How could the data in S3 be used by other applications beyond climateprediction.net? How would this cope with much bigger datasets of hundreds of terabytes or petabytes?

S3 can store files up to 5TB (as described on https://aws.amazon.com/s3/faqs/ ) so if datasets are larger than that another solutions should be explored (like a CephFS cluster, which is compatible with S3 via a Gateway API). With S3 share can be simply done by using the built-in web server and access policies. This information is now included in the text.

A few typos have been detected...

Thanks, indicated typos have been fixed.